# Evolving Storytelling: Benchmarks and Methods for New Character Customization with Diffusion Models

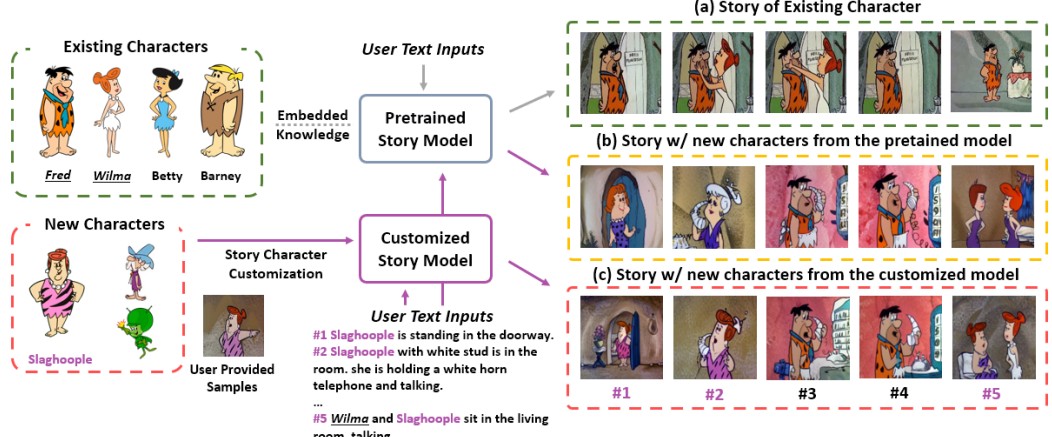

**Figure 1: Given a text-to-image story generation model trained on cartoons, we aim to customize the pretrained model so that end users can create new storylines featuring unseen new characters specified by only one example story (*i.e.*, 5 frames). For example, the model pretrained on *The Flintstones* is capable of generating visual stories for existing characters such as _Fred_ and _Wilma_, who are frequently depicted in the training datasets (row (a)). However, it falls short when generating stories featuring unseen new characters like Slaghoople because it has little prior knowledge of her (row (b)). Our customized model can take only one story of Slaghoople and generate new stories involving both new and existing characters (row (c)).**

## ABSTRACT

Diffusion-based models for story visualization have shown promise in generating content-coherent images for storytelling tasks. However, how to effectively integrate new characters into existing narratives while maintaining character consistency remains an open problem, particularly with limited data. Two major limitations hinder the progress: (1) the absence of a suitable benchmark due to potential character leakage and inconsistent text labeling, and (2) the challenge of distinguishing between new and old characters, leading to ambiguous results. To address these challenges, we introduce the NewEpisode benchmark, comprising refined datasets designed to evaluate generative models' adaptability in generating new stories with fresh characters using just a single example story. The refined dataset involves refined text prompts and eliminates character leakage. Additionally, to mitigate the character confusion of generated results, we propose EpicEvo, a method that customizes a diffusion-based visual story generation model with a single story featuring the new characters seamlessly integrating them into established character dynamics. EpicEvo introduces a novel adversarial character alignment module to align the generated images progressively in the diffusive process, with exemplar images of new characters, while applying knowledge distillation to prevent forgetting of characters and background details. Our evaluation quantitatively demonstrates that EpicEvo outperforms existing baselines on the NewEpisode benchmark, and qualitative studies confirm its superior customization of visual story generation in diffusion models. In summary, EpicEvo provides an effective way to incorporate new characters using only one example story, unlocking new possibilities for applications such as serialized cartoons.

## CCS CONCEPTS

• **Computing methodologies** → *Neural networks*; **Computer vision**; **Computer vision tasks**.

## KEYWORDS

Generative Diffusion Model, Story Visualization, Generative Model Customization

**ACM Reference Format:**
Anonymous Authors. 2024. Evolving Storytelling: Benchmarks and Methods for New Character Customization with Diffusion Models. In *Proceedings of the 32nd ACM International Conference on Multimedia (MM'24), October 28-November 1, 2024, Melbourne, Australia.* ACM, New York, NY, USA, 10 pages. https://doi.org/10.1145/nnnnnnn.nnnnnnn

# 1 INTRODUCTION

Diffusion models have demonstrated impressive performance in visual story generation tasks [19], holding promise for visual narratives, such as generating a new episode of existing comic books while maintaining the storyline continuity. However, existing generative models [19, 23–25, 27, 30] are highly limited when visualizing a story involving characters that appear infrequently or are absent from the training data. As illustrated in Fig. 1, existing visual story generation models struggle to depict *Slaghoople* (a character we held out from the training set) consistently, resulting in varied appearances due to the lack of character priors. Moreover, this generalization ability to unseen characters (referred to as *story character customization* in our paper) has not been adequately evaluated due to the absence of proper benchmarks in previous works [19, 23–25, 27, 30]. In this paper, **we systematically investigate story character customization by curating an appropriate dataset and designing a new method tailored to this task.**

**Our *NewEpisode* Benchmark**: To systematically address the challenge of customizing visual story generation models, we identified a critical issue: the absence of datasets featuring new characters in the testing split. To tackle this, we introduce the *NewEpisode* benchmark, whose test set contains unseen characters that have not appeared in the pretraining[1] set. Refining existing datasets [19, 23], we exclusively utilize stories with main characters for the pretraining and introduce minor supporting characters as new characters in the test set. It is worth noting that this process is not merely a re-dividing of existing datasets due to the ambiguous textual descriptions in these datasets. Minor characters are often referred to by a generic name in texts (*e.g.*, "an alien" instead of its character name), often causing the accidental inclusion of the unseen test characters in the pretraining set. To prevent this leakage, we manually investigated every single image-text pair of existing datasets, and even modified some textual descriptions to disambiguate the referent characters (see Fig. 2 for details). In contrast to datasets for general customization [3, 5, 15, 28, 35], which focus on personalizing a single concept and/or merging multiple concepts based on a pretrained general text-to-image model [33, 36], our NewEpisode benchmark is built for visual story generation models and offers testing samples that focus more on story character dynamics. This allows for in-depth exploration of how new characters can be integrated into existing storylines while avoiding disrupting the established character dynamics. NewEpisode benchmark thus provides a more nuanced approach to story character customization, emphasizing the seamless integration of new characters into complex narratives. We summarize some differences between NewEpisode and datasets used in other model customization works in Table. 1.

**Technical Challenges**: How do existing off-the-shelf methods [3, 15, 35] perform on our new dataset? Our preliminary studies indicate that existing methods might fall short within the context of story character customization. Specifically, we identify the core challenge is that: compared to a general text-to-image model [33], visual story generation models [27, 30] pretrained on a specific story dataset have stronger priors of existing characters. For instance, in Fig. 4-b, the model might be confused by the appearance of

---

[1]We use *pretraining* to refer to the process of training a visual story generation model and *customization* to refer to the process of adding new characters.

the lizard-like creature, *Rockzilla*. As a result, the new character might be straightly ignored, such as the case in Fig. 2-h, or the new character might be rendered in a corrupted way, such as the case in Fig.2-g.

**Our *EpicEvo* Model**: To address the aforementioned challenge, we propose a story character customization method, namely *EpicEvo*, which is featured in Fig. 3. In order to mitigate the overly strong influence of existing characters, we design an adversarial character alignment mechanism. As the name suggests, this mechanism is inspired by GANs[4], but we have devised it to be compatible with modern diffusion models. Specifically, we train a discriminator to judge whether the images generated by the visual story generation network contain a certain character in the latent space of the final step of the diffusion process. Our experiments show that this alignment encourages the generation of both new and existing characters whenever they appear in the captions. Additionally, we empirically found that the diversity of output stories can decrease if the model is extensively tuned on the customization samples. To mitigate this decline in diversity, we introduce a distillation loss, which effectively leverages both the pretrained model and customized models to enhance the generation quality even further.

In summary, our main contributions are threefold:

(1) We propose the *NewEpisode* benchmark, which contains refined datasets for pertaining visual story generation networks and a group of new characters available for the training and testing of story character customization methods. The NewEpisode benchmark presents a non-trivial challenge to existing visual story generation models and customization methods.

(2) We introduce *EpicEvo*, our story character customization method that enables the evolution of existing storylines by allowing the model to learn to generate stories featuring existing and/or new characters. EpicEvo encourages the model to generate characters distinctively using the adversarial character alignment module and it preserves model priors via distilling knowledge from a pretrained model.

(3) To the best of our knowledge, NewEpisode is the first benchmark specifically built for story character customization within the context of visual story generation. Our method, EpicEvo, is designed to tackle this challenging benchmark such that it enables end-users to create their branch of new stories with new characters using a few samples, making it possible to continue the legend described by the canon.

# 2 RELATED WORKS

**Text-to-image Generation** has been a heated topic in recent years. Earlier generative models were mostly based on GAN [4]. Follow-up works [1, 10, 11, 40] based on GAN [4] improved the model complexity and led to noticeable improvements in the synthesized image quality. Some other technical approaches featuring VAE [14] and VQ-VAE [32] also demonstrated the abilities to generate realistic images. Later, diffusion-based models [2, 8, 26, 31, 36] emerged as promising ways of composing more complex and diverse images based on text prompts. Recently, the emergence of latent diffusion models, *e.g.*, Stable Diffusion [33], demonstrated unprecedented

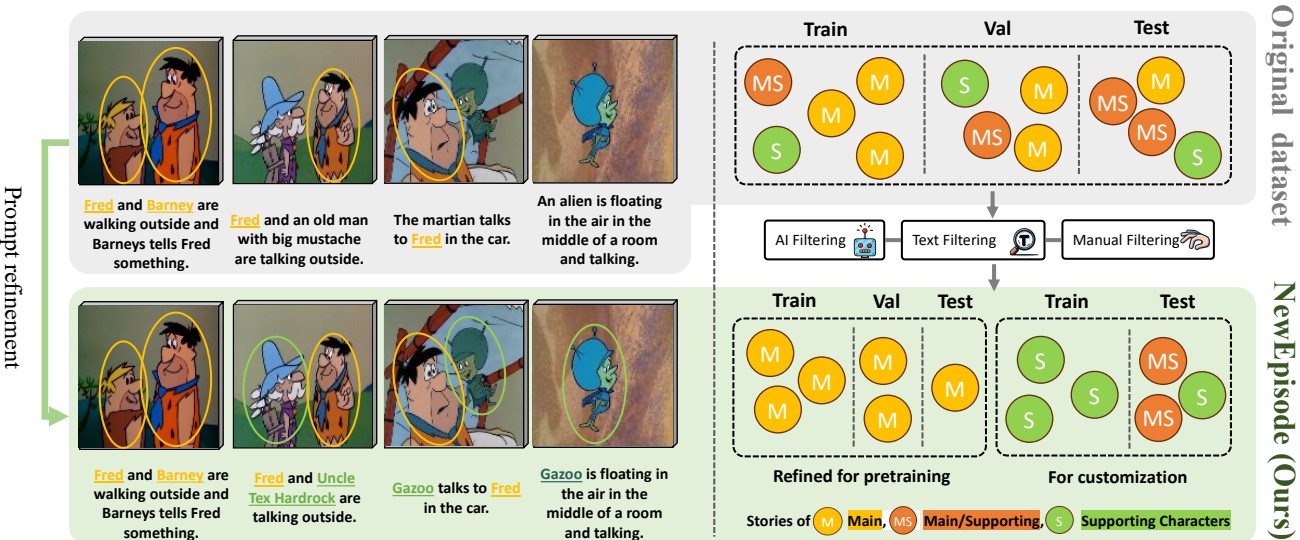

**Figure 2: Illustration of the dataset construction. The proposed dataset has two main contributions compared with the original datasets (FlintStonesSV [23] and PororoSV [19]): first, as shown on the left side, the original text prompts lack description for supporting characters, making the training/evaluation less tractable while our dataset provides more comprehensive and consistent annotations. Second, the original data can not well establish the adapting performance of models on new characters since "the leakage of character information" is shown on the right side. We reorganize the dataset based on our previously more detailed annotated dataset so that there is no leakage of new characters in the customization set to the pertaining set.**

generative ability by performing diffusion operations in the latent space of a VAE encoder [14].

**Visual Story Generation** involves creating sequences of images that form a coherent visual narrative, *i.e.*, a series of images with consistent contents such as characters, objects, background, style, etc. StoryGAN [19] first introduced the concept of visual story generation. [19] synthesizes several images based on the same number of prompts. The model processes contexts, such as prompts for all images, to generate more coherent content. Following works [17, 23, 24] further enhanced the model capability based on other techniques. With the rise of latent diffusion models [33], [25, 30] propose to synthesize visual stories based on pretrained diffusion models [33]. Specifically, Make-A-Story [30] constructs a memory module to store historical information such as previously generated images and prompts and utilize such information to condition the latent diffusion model [33]. Another work [27] proposes a more complex architecture that leverages multi-modal encoders [18, 29] to better encode historical information. Recently, [20] proposed the task of open-ended story generation where the task is to generate a series of images with 1-2 characters recurring. But it falls short for story with more complex dynamics, such as dynamics between characters. In sum, visual story generation models are good at generating stories for characters it has frequently seen during training, but struggle to generalize for new characters.

**Model Customization** enables end users to generate images that contain unique concepts, *e.g.*, an object, a certain art style. Previous model customization methods [3, 15, 35] primarily focus on customizing an off-the-shelf text-to-image model, such as Stable Diffusion [33]. Specifically, [35] and [15] train the text-to-image model to associate a unique concept with an uncommon token, and [3] tries to invert the text embedding vector that could prompt the model to generate a unique concept. Following works [5, 28] further combines customization with efficient adaptation methods [9] and proposes generating multiple concepts in one image. Still, our preliminary research finds efficient tuning methods [9] or tuning a smaller portion of the diffusion model can be less efficient for the story character customization task. In this paper, we focus on customizing visual story-generation models such that the customized model could generate stories for new characters.

## 3 DATASET AND BENCHMARK

In this section, we illustrate how the *NewEpisode* benchmark is constructed. For simplicity, without further notice, a *story* is assumed to have 5 images and 5 corresponding text prompts. Each image can feature different characters and images can re-appear in multiple stories.

### 3.1 Dataset

**NewEpisode**$_{FlintStones}$. The proposed dataset NewEpisode$_{FlintStones}$ is based on FlintStonesSv [23] dataset that originated from a text-to-video dataset proposed by [6]. The authors of [23] sampled images from consecutive video clips in the text-to-video dataset and there are 20132, 2071, and 2309 samples for training, validation, and testing. Each story contains a fixed length of 5 images and 5 corresponding text captions. The occurrence of seven main characters is annotated in the text captions, and there is a group of supporting

**Table 1: Differences between NewEpisode and other general model customization datasets. NewEpisode focuses on customizing a visual story generation model with new characters filtered from the original story visualization datasets. To ensure content-consistent storytelling, the underlying model of our method is a Visual Story Generation model [27] tuned on the pretraining datasets of the NewEpisode instead of a general stable diffusion model.**

| Method | Task | Backbone | New Concept Types | Number of New Concepts | Scale of Evaluation |
|---|---|---|---|---|---|
| DreamBooth [35] | | SD [33] / Imagen [36] | Daily Objects, Pets | 30 | 5 |
| Textual Inversion [3] | | SD [33] | Daily Objects, Pets, Humans, Styles | N/A | N/A |
| Custom Diffusion [15] | Single Image Generation | SD [33] | Daily Objects, Pets, Cars, etc. | 101 | 10 |
| Mix of Show [5] | | SD [33] | Movie Characters, Cartoon Characters | N/A | N/A |
| Orthaganol Adaptation [28] | | SD [33] | Movie Characters, Pets | 12 | 16 |
| **Ours (NewEpisode)** | Visual Story Generation | Visual Story Generation Model [27] | Cartoon Characters | 15 | 40 / 200 |

characters only appearing a few times without being carefully annotated as shown in Fig. 2. To create datasets free of these supporting characters for pretraining and customization, we manually went through the entire dataset with a supporting LLM tool [21]. In sum, we identify nine supporting characters in the original training, validation, and testing split. There are 382 different images containing these characters and 829 stories containing these new characters (the same image can appear in different stories). We construct the customization dataset based on these stories that depict interactions between these new characters and existing characters. We take one story for each new character during customization and subsequently test the customized model on all other stories about these new characters.

**NewEpisode**$_{Pororo}$. Similarly, PororoSV [19] is also sampled from a dataset originally used for other purposes [12]. In the PororoSV dataset, each story also contains 5 images and 5 corresponding text captions illustrating the character actions. There are 10191, 2334, and 2208 stories for training, validation, and testing. Upon close inspection, we found the captions for PororoSV [19] are more organized, *i.e.*, supporting characters are also referred to by their names. Therefore, we performed a rule-based text matching for the text captions of each story and identified 2976 images related to these supporting characters and a total of 4976 stories that contain these images. Excluding them from the training split, we obtain the pretraining datasets for NewEpisode$_{Pororo}$ that do not contain these supporting characters, *i.e.*, new characters. For customization and testing the customized model, we follow the same protocol for NewEpisode$_{Flintstones}$.

## 3.2 Benchmarking

As illustrated in Table 1, we have a larger scale of evaluation stories available for quantitative and qualitative evaluation of the customized visual story generation model. We show the average number of available individual images for each character. During customization, the model is provided with 1 story for each new character (such stories are ensured to have images all relevant to the new character for efficient customization). For NewEpisode$_{Flintstones}$, our evaluation protocol lets the model generate all the relevant stories for each new character. For NewEpisode$_{Pororo}$, we randomly select a maximum of 100 stories for each new character using a fixed random seed as many images are reused multiple times in many stories. For quantitative evaluation, we focus on the images that contain the new character and calculate automatic metrics such as Fréchet inception distance (FID) [7], CLIP-T, and CLIP-I [29] for them. Specifically, we leverage the FID score as we have a larger

scale of samples compared to general model customization works. Meanwhile, we also evaluate the CLIP-I and CLIP-T scores as it is widely used in various model customization works [3, 5, 15, 28, 35].

## 4 METHODOLOGY

Given $M$ lines of text prompts $S_{txt} = \{L^1, ..., L^M\}$, the visual story generation task aims to generate $M$ images $S_{img} = \{I^1, ..., I^M\}$ based on these prompts. Standard text-to-image models could handle this task by generating each image in $S_{img}$ based on each text in $S_{txt}$. However, this can overlook the contextual information contained in other prompts as well as previously generated images. Thus, the visual story generation method [25, 27, 30] instead utilizes all text prompts and previously generated images to guide the underlying diffusion model. In other words, the $i$-th image is conditioned as $P(I^i|L^1, ..., L^M, I^1, ..., I^{i-1})$. Notably, previous works assume the model only needs to learn to generate stories that contain a closed set of characters. This is reflected during the dataset construction process as only these characters will be referred to by their names consistently. For supporting characters that appear much less frequently, they are referred to in various different ways. In this paper, we instead assume the model will first learn to generate visual stories with a close set of $N$ characters, *i.e.*, $C_{ext} = \{c_1, c_2, ..., c_N\}$. We regard this process as pretraining as the model requires a considerable amount of iterations to learn to generate coherent images by considering contextual information. Subsequently, in order to generate stories with new characters, we customize the visual story generation model to learn to generate content related to a group of new characters $C_{new} = \{c_{N+1}, ..., c_{N+K}\}$. We consider the customization process a few-shot finetuning process, *i.e.*, the model learns to generate an image containing the new character by only taking **a single story** about the new characters. Notably, we use finetuning stories that contain only the new characters themselves, which is closer to real-world cases where the end users cannot provide versatile images of the new characters, *e.g.*, how they interact with other characters.

## 4.1 Visual Story Generation Preliminaries

**Diffusion Models** approximate a sample distribution $p(x)$ by denoising from a base distribution in multiple steps. To learn such a model with parameter $\theta$, we define the forward diffusion process as adding noise to an image $x_0 \sim p(x)$ sampled from the sample distribution $p(x)$ following a Markov process for $T$ times. In practice, most recent works operate in the latent space of a VAE encoder [14] for various benefits [33]. Thus, we denote the VAE-encoded $x$ as $z$,

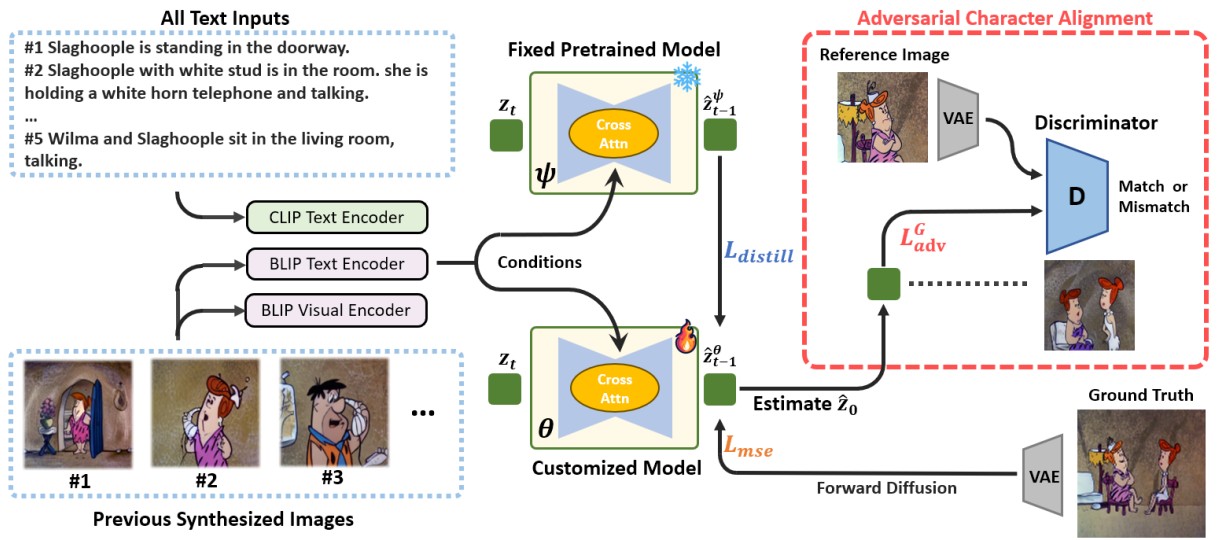

Figure 3: Illustration of EpicEvo. When generating the $i$-th image, the model takes all text inputs and previously generated images, encodes them using CLIP [29] and BLIP [18] text and visual encoders, and conditions the denoising network [34]. To enable better story character customization, the denoising network has three training objectives: 1) predict the noise $\epsilon$ added to noisy latent $z_t$ at time step t such that the estimated noise $\epsilon_\theta$ is close to the ground truth noise $\epsilon$. This is reflected by the mean square loss (MSE) loss $\mathcal{L}_{mse}$; 2) maximizing the probability that the discriminator network will classify the generated image as a matching image w.r.t. to the reference image, *i.e.*, minimize the adversarial character alignment loss $\mathcal{L}_{adv}^G$; 3) aligning with the pretrained model by minimizing a distillation loss $\mathcal{L}_{distill}$ calculated as the L2 distance between the noises estimated by the pretrained model and the customized model. We denote the latent denoised by the customized model and pretrained model as $\hat{z}_{t-1}^\theta$ and $\hat{z}_{t-1}^\psi$. Notably, both the diffusion process and adversarial alignment process operate in the latent space of the VAE [14] network. Thus, for the adversarial alignment process, we estimate $\hat{z}_0$ instead of directly computing $\hat{z}_0$ as this is too computationally expensive. The dotted line here indicates that the estimated $\hat{z}_0$ can be decoded as a prediction of the generated image in the pixel space. We omit the complete process of adding noise to the image to obtain $z_t$ and the iterative nature of the reverse diffusion process for simplicity. Best viewed in color.

and the forward and backward diffusion process can be described as follows:

$$q(z_{1:T}|z_0) = \prod_{i=1}^{T} \mathcal{N}(z_t; \sqrt{1-\beta_t}z_{t-1}, \beta_t I)$$

$$p_\theta(z_{0:T}) = p_\theta(z_T) \prod_{i=1}^{T} p(z_{t-1}|z_t), \qquad (1)$$

where $\{\beta_t \in (0,1)\}_{t=1}^{T}$ is a predefined time-dependent variance schedule. The network is learned to estimate the noise $\epsilon$ added at each time step $t$ to reconstruct $z_0$ progressively and this can be described as minimizing the L2 distance between the ground truth noise $\epsilon$ and estimated noise $\epsilon_\theta$ by the denoising network:

$$\mathcal{L}_{mse} = \mathbb{E}_{\epsilon,z,t} \left[ \|\epsilon - \epsilon_\theta(z_t, s, t)\|_2^2 \right], \qquad (2)$$

where $\epsilon \sim \mathcal{N}(0,1)$. $s$ refers to a conditioning vector that makes the generation process conditional, *i.e.*, the generation process is dependent on the input conditions, such as text, images, or information from other modalities.

**Diffusion-based Visual Story Generation Models** [27, 30] that achieved state-of-the-art performance recently are based on the latent diffusion model described above. The main difference

is that instead of only encoding a single line of prompt $L$ as the condition $s$, *i.e.*,

$$s = \mathcal{E}_{txt}(L), \qquad (3)$$

visual story generation models encode all text prompts in $S_{txt}$ and all generated images $S_{img}^i = \{I_1, ..., I_{i-1}\}$ before the $i$-th image as the condition. Formally, this can be described as:

$$s_i = F \left( \mathcal{E}_{txt}(S_{txt}), \mathcal{E}_{img}(S_{img}^i) \right), \qquad (4)$$

where $\mathcal{E}_{txt}$ and $\mathcal{E}_{img}$ are text and visual encoders, $s_i$ is the conditioning vector for the $i$-th image to be generated,$F$ is a function that fuses the encoded text and visual information.

## 4.2 Story Character Customization

**Pretraining.** Existing customization methods [3, 5, 15, 28, 35] aim to customize a general text-to-image model [33]. For visual story generation models [27, 30], the model needs to be first trained on a story dataset, such as the FlintStonesSV [23] dataset. Such a dataset contains a large number of stories that have a fixed length of 5 images and 5 lines of text captions. In the context of story character customization, we regard this process as pretraining. Notably, previous models are trained using a dataset that contains both $C_{ext}$

and $C_{new}$, whereas we train the underlying visual story generation network using the refined pretraining splits of NewEpisode and these splits are free of $C_{new}$.

**Customization.** Common customization methods customize a text-to-image model by tuning the model to generate unique concepts whenever prompted by a special word, or more precisely, a special token [3, 27, 35]. Within the context of story character customization, we customize the visual story generation model by having the model generate a new character whenever such a character is mentioned by the prompt. Specifically, we refer to new characters with a short phrase of descriptive words. We found this sufficient to prompt the model to generate stories containing these new characters in the case of story character customization. We use such a prompting strategy for our method, *i.e.*, EpicEvo, in the rest of this paper. Additionally, we also leverage a small group of stories of existing characters as regularization terms to mitigate overfitting. The optimization goal can be therefore formulated as minimizing the following loss:

$$\mathcal{L}_{mse} = \mathbb{E}_{\epsilon, z', t} \left[ \| \epsilon - \epsilon_\theta(z'_t, s, t) \|_2^2 \right] + \mathbb{E}_{\epsilon, z, t} \left[ \| \epsilon - \epsilon_\theta(z_t, s, t) \|_2^2 \right], \quad (5)$$

where $z'$ corresponds to stories of existing characters and $z$ corresponds to stories of new characters.

## 4.3 Adversarial Character Alignment

A previous study [15] alleviates the challenge of multi-concept customization through either naive joint training or merging model weights trained on individual concepts. Story character customization poses more difficulties due to the similarities between new and existing characters. In this work, we jointly train with multiple new characters. We noticed that the customized model might be unable to capture the visual traits of the new characters. For example, in Fig. 4-g, the model can be confused about the appearance of the new character, leading to unsatisfactory generation results.

In view of this, inspired by the adversarial learning scheme featured by [37], we propose the adversarial character alignment module designed for the diffusion-based visual story generation model. The key objective of the adversarial finetuning process is to regulate the model such that it generates each character distinctively and mitigates the confusion between each existing and new character. Following the common formulation of adversarial learning [4], we regard the diffusion model as the generator and we construct a discriminator $D_\phi$ with parameters $\phi$ to discriminate between the ground truth latent $z_0$ and the estimated latent $\hat{z}_0$. The key to enforcing better character alignment lies in the way we construct the positive and negative samples for the discriminator, that is:

- We first select several visual references for a certain character and encode them as $z_r$.
- We generate the positive samples by fusing the ground truth latent $z_0$ with the latent $z_r$ of the reference character whenever a certain frame in the story contains the reference character.
- We generate the negative samples by fusing the estimated latent $\hat{z}_0$ and the latent $z_r$ of the reference character whenever a certain frame in the story contains the reference character.

This forces the network to generate images containing the reference character. In practice, the fusion process is implemented as concatenation. The noisy latent $z_t$ is derived from a group of ground truth images $z_0$, *i.e.*, $z_t = \alpha_t z_0 + \sigma_t \epsilon$ where $\alpha_t = 1 - \beta_t$. The generated data from the model could be denoted as $\hat{z}_0$. Therefore, the optimization objective for the denoising network can be derived as:

$$\mathcal{L} = \mathcal{L}_{mse} + \lambda \mathcal{L}_{adv}^G(\hat{z}_0, z_r, \phi), \quad (6)$$

where $\lambda_1$ is the coefficient for the adversarial loss for the diffusion network. In practice, we set the optimization goal for the generation as minimizing the following loss:

$$\mathcal{L}_{adv}^G = -\mathbb{E}_{\hat{z}_0, z_r} \left[ \log(D_\phi(\hat{z}_0, z_r)) \right] \quad (7)$$

whereas the discriminator is trained to maximize:

$$\mathcal{L}_{adv}^D = \mathbb{E}_{z_0, z_r} \left[ \log(D_\phi(z_0, z_r)) \right] + \mathbb{E}_{\hat{z}_0, z_r} \left[ \log(1 - D_\phi(\hat{z}_0, z_r)) \right] \quad (8)$$

In practice, the discriminator $D_\phi$ contains 4 convolutional layers and 1 linear layer to process the positive and negative samples and it operates in the latent space instead of the pixel space. Instead of stepping the model from the $t$-th time step to 0 to obtain $\hat{z}_0$, an alternative way is to substitute $z_0$ based on [38], *i.e.*,

$$\tilde{z}_0 = \frac{x_t - \sqrt{1 - \overline{\alpha}_t} \cdot \epsilon_t}{\sqrt{\overline{\alpha}_t}}, \quad (9)$$

where $\overline{\alpha}_t = \prod_{i=1}^T \alpha_i$, and $\tilde{z}_0$ is an estimated version of $z_0$. Intuitively, the generator network, *i.e.*, the diffusion model, generates images that contain the reference character such that it maximizes the likelihood that these images are deemed a 'match' by the discriminator, while the discriminator tries to distinguish whether the generated images contain the reference character.

## 4.4 Story Prior Preservation via Distillation

Finetuning based on a few training samples introduces risks of overfitting, resulting in a decrease in diversity and many other undesirable outcomes such as language drift [15, 16, 22, 35]. Within the context of visual story generation, we also observe that the model could associate characters with certain types of backgrounds, further decreasing the output diversity. Meanwhile, the pretrained network possesses a higher level of output diversity despite it has little knowledge of the new characters prior, making them suboptimal for directly generating these characters. Therefore, we design a distillation loss where the pretrained story generation model with parameter $\psi$ is regarded as a teacher and the distillation loss is formulated as:

$$\mathcal{L}_{distill} = \mathbb{E}_{z_0, s, t} \left[ \| \epsilon_\psi(z_t, s, t) - \epsilon_\theta(z_t, s, t) \|_2^2 \right] \quad (10)$$

To sum up, the overall optimization objective for the story visualization network can be formulated as:

$$\mathcal{L} = \mathcal{L}_{mse} + \lambda_1 \mathcal{L}_{adv}^G + \lambda_2 \mathcal{L}_{distill}, \quad (11)$$

where $\lambda_1, \lambda_2$ are two coefficients for the generator adversarial loss and distillation loss.

**Table 2: FID, CLIP-I, and CLIP-T scores on the FlintStones and Pororo split of the NewEpisode benchmark**

| Methods | NewEpisode$_{FlintStones}$ | | | NewEpisode$_{Pororo}$ | | |
|---|---|---|---|---|---|---|
| | FID↓ | CLIP-I ↑ | CLIP-T ↑ | FID↓ | CLIP-I ↑ | CLIP-T ↑ |
| Pretrain Model | 210.45 | 0.8203 | 0.2510 | 143.98 | 0.7690 | 0.2358 |
| Textual Inversion | 207.36 | 0.8027 | 0.2399 | 150.08 | 0.7603 | 0.2335 |
| DreamBooth | 192.95 | 0.8360 | 0.2510 | 134.94 | **0.8010** | 0.2522 |
| Custom Diffusion | 190.97 | 0.8250 | 0.2480 | 131.88 | 0.7935 | 0.2487 |
| **EpicEvo** | **188.30** | **0.8380** | **0.2573** | **130.4** | 0.7954 | **0.2551** |

## 5 EXPERIMENTS

In this section, we describe the details of the NewEpisode benchmark, followed by implementation details of baselines and our method, quantitative analysis, and qualitative results.

**Dataset.** We collected a dataset that contains thousands of visual stories for a total of 15 new characters from the FlintStonesSV [23] and PororoSV [19] dataset. Readers can refer to Sec. 3, Fig. 2, and Table 1 for a detailed illustration of how we derived our dataset from existing datasets and benchmark model customization methods.

**Training Details.** We use the model proposed by [27] as the backbone for visual story generation. During the pretraining stage, we keep most of the settings the same as [27] but train the network at a resolution of $256 \times 256$ as the training samples from FlintStonesSV [23] and PororoSV [19] are $128 \times 128$. During model customization, we freeze the CLIP [29] and BLIP [18] encoders and only train the denoising network. The batchsize during customization is 2 and the learning rate is $1 \times 10^{-5}$. The model is tuned for 100 epochs using the Adam [13] optimizer. For sampling, we step the model for 50 steps using the DDIM [38] sampler with a guidance scale of 6.0. $\lambda_1$, $\lambda_2$ is set to 0.75, 0.5 and 0.25, 0.25 for the Flintstones and Pororo customization benchmark in NewEpisode. We consider multiple model customization works [3, 15, 35] as competitive baselines. For DreamBooth [35], we randomly select rare tokens and finetune the entire model following the original paper. Since we use a small portion of training samples during customization, we disable the prior-preservation loss of DreamBooth [35]. For Custom Diffusion, we employ the same group of tokens as DreamBooth while following its experiment setup. For Textual Inversion [3], we initialize new token embedding using the embedding mean and variance.

**Evaluation Metrics**. To quantitatively evaluate our method and various baselines, we employ three widely adapted metrics, including Fréchet inception distance (FID) [7], CLIP-I, and CLIP-T [15, 29, 35]. CLIP-I refers to image-to-image similarity score and CLIP-T refers to text-to-image similarity score. Specifically, we focus on the images related to new characters (not every image in one story contains the new characters) the FID score measures the cluster distance between the generated images and ground truth in the latent space of the Inception V3 model [39]. We average across the FID score for each character and report the average character FID. For CLIP-based [29] metrics, CLIP-I is the average pairwise cosine similarity between the CLIP features of generated images and ground truth images. CLIP-T is the average pairwise cosine similarity between the CLIP features of generated images and the text captions. When calculating the CLIP-T score, we use descriptive phrases to refer to each new character because the modifier tokens

**Table 3: Ablation studies for the distillation process and adversarial character alignment process**

| Methods | NewEpisode$_{FlintStones}$ | | | NewEpisode$_{Pororo}$ | | |
|---|---|---|---|---|---|---|
| | FID↓ | CLIP-I ↑ | CLIP-T ↑ | FID↓ | CLIP-I ↑ | CLIP-T ↑ |
| Pretrained Model | 210.45 | 0.8203 | 0.2510 | 143.98 | 0.7690 | 0.2358 |
| $\mathcal{L}_{mse}$ | 192.76 | 0.8360 | 0.2542 | 132.51 | 0.7940 | 0.2546 |
| $\mathcal{L}_{mse} + \mathcal{L}_{distill}$ | 192.04 | 0.8345 | 0.2550 | 130.73 | 0.7915 | 0.2537 |
| $\mathcal{L}_{mse} + \mathcal{L}_{distill} + \mathcal{L}_{adv}^{G}$ | 188.30 | 0.8380 | 0.2573 | 130.40 | 0.7954 | 0.2551 |

and inverted tokens are out-of-distribution for the pretrained CLIP [29] model.

**Quantitative Analysis**. We show the FID scores in Table. 2. Notably, the scale of the FID score highly relies on the sample size, with a smaller sample size, it is expected to have larger FID scores. Nevertheless, the reduction in FID scores on both NewEpisode$_{Flintstones}$ and NewEpisode$_{Pororo}$ indicate that our story character customization method, i.e., EpicEvo, could on average achieve better customization results compared to the baselines. In terms of the CLIP-I and CLIP-T scores, we found our model achieves a higher CLIP-I score on the NewEpisode$_{Flintstones}$ and our method reaches the second place on the test set of NewEpisode$_{Pororo}$. Still, for the CLIP-T score, our model obtains better similarities compared to the baselines. This indicates the generated stories of new characters are more semantically aligned with the text prompts. The results of Textual Inversion [3] and the pretrained model validate that the model has never seen these new characters, making it hard for them to generate stories related to these characters.

**Ablation Study**. We ablate our method, EpicEvo, by removing three losses, i.e., the reconstruction loss $\mathcal{L}_{mse}$, the adversarial character alignment loss $\mathcal{L}_{adv}$, and the distillation loss $\mathcal{L}_{distill}$. Starting from a pretrained model trained on the pretraining split of NewEpisode, we validate the model performance using three of our testing metrics. By finetuning the model using a few examples of each new character, we notice a significant drop in terms of the FID score, meaning the model is learning to generate content more relevant to the new characters. Next, enabling the distillation loss decreases the FID score but increases the CLIP-I and CLIP-T scores. We empirically found the reason might be the pretrained model has no prior of the new character. Therefore, learning from the pretrained model has the risk of misguiding the customized model despite it could better prevent overfitting. We recommend using a smaller $\lambda_2$ such that the distillation loss could encourage the model to generate diverse contents without disturbing learning of the visual features of new characters. Lastly, we enable the character alignment loss we designed for the diffusion-based visual story generation model, i.e., $\mathcal{L}_{adv}^{G}$. We observe a further reduction in the FID score and an increase in the CLIP-I and CLIP-T scores, validating that the proposed adversarial character alignment method could encourage the model to learn to generate new characters more consistently.

**Qualitative Analysis**. While our method reached better customization performances on both customization testing datasets of the NewEpisode, it is still an open debate regarding to what extent FID score [7], CLIP-I and CLIP-T [29], etc. can represent human perceptions. . Thus, to empirically validate the performance of EpicEvo, we display the generated images of new characters in Fig. 4, and

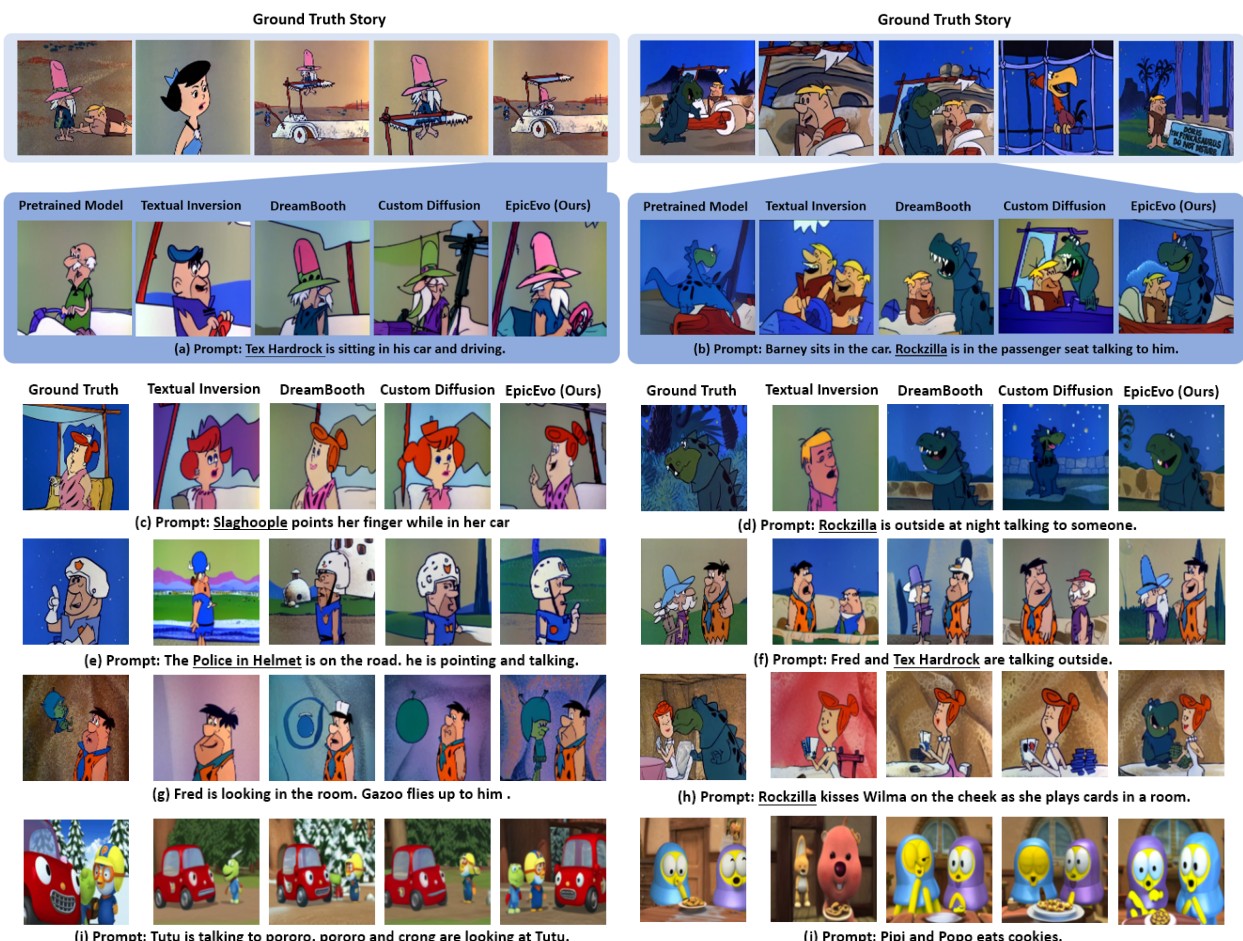

**Figure 4: Qualitative story character customization results from different baselines. For (a) and (b), we highlight the original visual story and highlight one of the generated frames containing the new character. For the rest of the image, we present sampled individual frames from stories to better demonstrate the effectiveness of EpicEvo under various conditions.**

we show stories and images relevant to the new characters in Fig. 4. Overall, Fig. 4 demonstrates that our method can generate the new characters alone, *e.g.*, Fig. 4-a,b,c,d,e or with other characters, *e.g.*, Fig. 4-f,g,h,i,j more consistently. We also find Dreambooth [35] achieves decent results, specifically on NewEpisode_Pororo. We hypothesized that the Custom Diffusion [15] is limited because simply tuning the cross-attention layer might not be sufficient to learn the novel appearance of new characters, this also aligns with our preliminary studies that show LoRA [9] offers unsatisfactory results due to its limited ability to learn representations. For Textual Inversion [3], users rarely pick images generated by it, and this is expected as the model will struggle to invert concepts it has never seen before, especially under the condition that we carefully removed all these new characters from the pertaining data.

## 6 CONCLUSION

In this paper, we tackle the challenging problem of story character customization. We aim to customize a visual story generation model so that it can generate stories for new characters it has never seen before. We first propose the NewEpisode benchmark. NewEpisode leverages supporting characters in previous story generation datasets as new characters. It contains carefully refined pretraining data to train visual story generation models and plenty of data for training and testing story character customization methods. We identify the core challenge is that visual story generation contains complex priors such as character dynamics, making customizing more challenging. In view of this, we propose EpicEvo, our method to tackle story character customization. EpicEvo takes a few images of the new character and customizes the model to generate stories for the new character. It contains an interesting adversarial character alignment module and it utilizes knowledge distillation to prevent overfitting. Compared to previous methods, we quantitatively and qualitatively validate that EpicEvo can generate visual stories that have better new character consistency. This indicates that EpicEvo can better tackle the problem of story character customization, making downstream tasks such as the creation of serialized cartoons, and TV series, possible.

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
