# OpenReview forum: "Evolving Storytelling: Benchmarks and Methods for New Character Customization with Diffusion Models"
_acmmm.org/ACMMM/2024/Conference — MM2024 Oral_

### Official Review · Reviewer_3otk · 2024-05-19

**Rating:** 4
**Confidence:** 3

**Summary:**

This paper introduces a new task termed story character customization, which aims to generate stories for new characters the model has never seen before. To this end, NewEpisode benchmarks based on FlintStonesSV and PororoSV datasets are developed. Besides, an elaborate model EpicEvo is designed. This model uses an adversarial character alignment module to better distinguish new characters and a distillation module to prevent overfitting. Extensive experiments validate the effectiveness of the method.

**Strengths:**

1.	The story character customization task is novel and interesting. I believe it will have a wide range of applications.
2.	The experiment is basically sufficient. Both quantitative and qualitative results (including user studies) are included to demonstrate the effectiveness of the method.
3.	In the supplementary, the authors analyze the limitations of this work, which is insightful.

**Limitations:**

1.	In Line 364, the authors said going through the FlintStones dataset with a supporting LLM tool. How to implement it in detail?
2.	As illustrated in Line 629, I wonder how to select the visual references? And does this lead to potential unfairness?
3.	The adopted evaluation metrics (FID, CLIP-I and CLIP-T) tend to measure the image quality or the consistency between captions and images. However, as far as I know, the character F1-score is a common metric used in previous work. I think this metric should also be considered in this work.

**Suitability:**

3

---

### Official Review · Reviewer_N2Qb · 2024-05-19

**Rating:** 5
**Confidence:** 2

**Summary:**

This paper introduces the NewEpisode benchmark and the EpicEvo method for customizing visual story generation models to include new characters. The NewEpisode benchmark is designed to evaluate the adaptability of generative models in generating new stories with new characters using a single example story. The benchmark includes datasets derived from FlintStonesSV and PororoSV, focusing on the seamless integration of new characters into established narratives. EpicEvo incorporates an adversarial character alignment module and knowledge distillation to align generated images progressively with exemplar images of new characters while preventing forgetting of existing characters and background details. The method demonstrates superior performance in both quantitative and qualitative evaluations compared to existing baselines.

**Strengths:**

1. The introduction of the NewEpisode benchmark is a significant contribution to the field, providing a structured way to evaluate story character customization in visual story generation models.

2. The EpicEvo method's adversarial character alignment module is an innovative approach to ensuring the consistency of new characters within generated stories. This method helps mitigate the confusion between new and existing characters.

3. The work has practical implications for the creation of serialized cartoons and TV series, showcasing the potential for real-world applications of the proposed method.

4. The availability of the dataset and code enhances the reproducibility and practical utility of the research, making it accessible for further exploration and development by the research community.

**Limitations:**

1. The method and benchmark are tailored specifically for visual story generation models. The applicability of these findings to other types of generative models remains to be explored.

2. Despite the use of knowledge distillation, there is still a risk of overfitting during the customization process, which could impact the diversity and quality of the generated stories.

3. From the results shown in Fig. 4, it is evident that the generated characters sometimes exhibit inconsistencies in details and proportions.  How do the authors plan to address the inconsistencies in character details and proportions, and what future steps are envisaged to enhance the method’s robustness and applicability?

**Suitability:**

3

---

### Official Review · Reviewer_qW9y · 2024-05-24

**Rating:** 4
**Confidence:** 3

**Summary:**

This paper studies new character customization for story visualization. It first introduces the NewEpisode benchmark by refining existing datasets FlintStonesSV and PororoSV. The novel EpicEvo is then proposed to customize a diffusion-based visual story generation model. The experiment results verify the effectiveness of the proposed approach.

**Strengths:**

1. The proposed benchmark benefits the community if the authors make it publicly available.
2. Motivation is simple and clear.

**Limitations:**

1. It's recommended to highlight the best results in Table 3 in bold.
2. The rudimentary dataset partitioning strategy inherently restricts such methodologies' applicability in a broader field. Fundamentally, the volume of the dataset remains finite.
3. The term ‘Storytelling’ in the title confuses me. From my understanding, this is a work on story visualization.

**Suitability:**

3

---

### Official Review · Reviewer_zUJ2 · 2024-05-26

**Rating:** 5
**Confidence:** 2

**Summary:**

This paper introduces the NewEpisode benchmark, comprising refined datasets designed for model evaluation on generating new stories with fresh characters. It also presents a story customization method for storytelling.

**Strengths:**

- The paper is well organized.
- The idea of reorganizing the dataset for the customization task and proposing the customization approach is somewhat innovative.

**Limitations:**

- The effectiveness of the distillation process has not been proven. In Table 3, when comparing the lines "L_mse+L_distill" and "L_mse," it appears that the introduction of L_distill negatively impacts performance.
- The main text lacks a qualitative comparison of generated stories between EpicEvo and other baseline methods.
- Typos:
1) Line 176-178: Fig. 2-h, Fig. 2-g;
2) Line 810: Two full stops.

**Suitability:**

3

---

### Meta-Review · Area_Chair_oLPo · 2024-07-10

**Recommendation:** Accept (Oral)
**Confidence:** 5

**Metareview:**

All the reviewers gave positive ratings and tend to accept the paper. SAC and AC agree with reviewers and recommend acceptance of the paper.